# Smart Node Networks Orchestration: A New E2E Approach for Analysis and Design for Agile 4.0 Implementation

**DOI:** 10.3390/s21051624

**Published:** 2021-02-26

**Authors:** Annalisa Bertoli, Andrea Cervo, Carlo Alberto Rosati, Cesare Fantuzzi

**Affiliations:** DISMI, University of Modena and Reggio Emilia, 41121 Modena, Italy; annalisa.bertoli@unimore.it (A.B.); andrea.cervo@unimore.it (A.C.); carloalberto.rosati@unimore.it (C.A.R.)

**Keywords:** cyber-physical systems, wireless sensor networks, multi-agent systems, decentralized automation systems, industrial internet of things, smart nodes control

## Abstract

The field of cyber-physical systems is a growing IT research area that addresses the deep integration of computing, communication and process control, possibly with humans in the loop. The goal of such area is to define modelling, controlling and programming methodologies for designing and managing complex mechatronics systems, also called industrial agents. Our research topic mainly focuses on the area of data mining and analysis by means of multi-agent orchestration of intelligent sensor nodes using internet protocols, providing also web-based HMI visualizations for data interpretability and analysis. Thanks to the rapid spreading of IoT systems, supported by modern and efficient telecommunication infrastructures and new decentralized control paradigms, the field of service-oriented programming finds new application in wireless sensor networks and microservices paradigm: we adopted such paradigm in the implementation of two different industrial use cases. Indeed, we expect a concrete and deep use of such technologies with 5G spreading. In the article, we describe the common software architectural pattern in IoT applications we used for the distributed smart sensors, providing also design and implementation details. In the use case section, the prototypes developed as proof of concept and the KPIs used for the system validation are described to provide a concrete solution overview.

## 1. Introduction

The field of cyber-physical systems is an industrial IT and automation research topic that addresses the deep integration of computing, communication and process control with humans in the loop (Figure 1), with the final goal to define modelling, controlling and programming methodologies for managing industrial mechatronics systems [1]. Cyber-physical systems are the key technology enabling Industry 4.0 [2] and can be applied on different levels in the modern value chain, meaning that it can be used to control and design the product for the end consumer, can be the supporting technology for process control behind the final product creation, can be the technology supporting the smart supply chain or, as recent research demonstrates, can be also a supporting tool to manage the end of life of modern products. The last topic needs special attention since historically there is a different velocity in technology development and EOF management development [3]; the interesting aspect with distributed systems, like cyber-physical systems, is that there is no more a monolithic and “time-constant” product, but the product definition is the temporary configuration and state of several atomic entities that in a certain time cooperate and participate in the definition of a more complex ecosystem (also defined holon [4]) perceived as entire/global system. For this reason, it is important to structure end-to-end life-cycle management systems with special attention on configuration management and centralized monitoring systems to tackle such complexity, focusing on legibility and smart human-machine interfaces. Moreover, such pervasive connectivity enables circular economy and smart supply chain systems, simplifying data exchange among different companies and systems involved in different tasks of the end-to-end process. Involved in such context there are operation, like distribution, waste management, recycling [5] but it can also be used as a retroaction to improve processes, like data analytics from the perspective of service provider (e.g., smart prognostic and diagnostic to guarantee continuity and continuous improvement of services for the clients).

Central characteristics of such systems are intelligence, self-awareness, self-management and self-configuration [6]. Such characteristics are also shared in terms of modelling and implementation with multi-agent systems that seems a promising approach to manage cyber-physical systems thanks to behaviour driven modelling and encapsulation where it is possible to represent different behaviours depending on different input triggers due to external technical systems, external virtual systems or interaction with human beings, creating in this way a human in the loop ecosystems [7]. An important area in this domain is data mining and orchestration of smart nodes. With smart nodes we define embedded systems having physical sensors as interface with real world and software services capable of implementing business logics and algorithms to filter data, expose them to other services for data analytics processes or to increase the shared knowledge of the system and apply event detection mechanisms to identify events of interest to be notified or managed. It is important that research community focus on both aspects enabling on the one hand smart nodes orchestration developing low power reliable sensor nodes with focus on sensor technologies and physics research, on the other hand developing the software infrastructure to collect filter, aggregate and analyse datasets in order to create new value for new application (decision support systems, data visualization and HMIs, automatic control systems, VR, AR, etc.) [8].

Modern industrial systems are so advanced that companies often struggle to exploit their potential, limiting themselves to a more proven use. There is no rigorous and unique way to deal with the distributed and deeply interconnected nature of modern industrial systems in a structured way, application independent.

In Section 2, we are going to explore the state of the art in terms of Industry 4.0 key technologies, the main obstacles for a wider adoption also in small-medium enterprises (SMEs) and main enablers for implementing industrial IoT and cyber-physical systems solutions.

In Section 3 we focus also on software technologies for distributed systems and IoT sensor networks, architectural patterns and HMI systems in order to integrate these complex systems in cooperative environment with human beings.

In Section 4 we describe the central role of universities for knowledge transfer and practical adoption of I 4.0 technologies in SMEs, describing the process with which we analysed stakeholders needs to be address with new enablers and the agile framework adopted to create awareness and competences growing. Then, we describe an organic approach derived from such experience in order to support companies with lack of specialized competences in IoT and SOAs architecture in introducing such new technologies in an easy and “plug-and-play oriented” way. To structure the analysis and product development flow we rely on a typical systems engineering swimlane, starting from functional requirements derived from initial needs identified with interviews, developing logical architecture and leveraging on multi-agents systems, IoT technologies, distributed software patterns and SOAs to achieve the design of the physical architecture [9,10]. Finally the validation process is explained. Currently the approach well investigates the way of working and technical implementation in case of wireless sensor networks (WSNs) and soft real-time requirements, mainly addressing the needs of easy implementation and cheap embedded systems: this represents the area of interest of this research and by consequence, the main limitation to be covered with future research.

In Section 5, we highlight the future research needs due to current limitations to achieve a broader scope (i.e., 5G opportunities, low power devices in the loop, devops for IoT), while the concluding remarks are given in Section 6.

## 2. State of the Art Review

### 2.1. Industry 4.0—General Concepts and Expectations

The Industry 4.0 concept was formalized for the first time in an article published in November 2011 by the German government; this article was part of an initiative regarding high-tech strategy for 2020 [11]. Nowadays the industry is moving to cyber-physical systems, and this change is desirable not only because of the new technology available and its evolution, but also by governments and their economic politics [12]. The development of information and communications technologies (ICT) triggered the fourth industrial revolution [13].

Compared to past revolutions, Industry 4.0 has some differences. First this industrial revolution was predicted, and this fact allowed companies to make some decisions to address this new industrial paradigm’s challenges. There is not a clear vision about this manufacturing paradigm, regarding its implications and consequences, and the challenges are not completely understood from the companies [11]; moreover, there are a lot of barriers that decrease the Industry 4.0 implementation [14]. As stated in [15] major barriers for the implementation of I 4.0 technologies in SMEs are represented by the technical background, the time to learn new methodologies and technologies and the costs: for these reasons with our paper we aim to propose both a state of the art review and a practical implementation guideline that demonstrate how to implement soft real-time monitoring systems via IoT, with affordable costs and implementation time.

As reported by Muller in [16], nowadays one of the principal barriers for full adoption of I 4.0 technologies is the lack of competencies, in particular lack of qualification in IT-related competencies (e.g., data analysis, streaming and manipulation with software); this creates the fear of job losses against the principles of I 4.0 cyber-physical cooperative systems. Another reason is the unclear comprehension of the new mansion requested in I 4.0 and the new role of the human operator [17].

### 2.2. Smart Technologies of Industry 4.0

Unlike the previous industrial revolutions, Industry 4.0 puts the individual customer’s needs as central objective. To accomplish this target, different areas are involved, such as order management, research and development, manufacturing commissioning, and delivery up to the utilization and recycling of products [18].

To achieve this goal, industries need a vertically and horizontally integrated production systems. Internet of things (IoT) and internet of service characterize the fourth industrial revolution, and they enable the evolution of smart factories. To manage this kind of systems, a large amount of data is required and collected, and exchange information and control processes in production and logistics become essential. The data are stored in a decentralized way to promote local decisions, but they are transparent to all the partners. To create this kind of environment, a cyber-physical system is required, because all the elements that compose it must be able to share information, act and control each other autonomously, avoiding failures [12]. Main technologies under the umbrella of Industry 4.0 are: cyber-physical systems, internet of things, collaborative robotics, virtual and augmented reality, cloud computing, industrial integration, enterprise architecture, service-oriented architectures, also defined industrial multi-agent systems [19]. There are some pillars in Industry 4.0 that will allow the transformation of isolated nodes into integrated and automated production flow, to optimize not only the singular node but the entire system. 

### 2.3. Industrial Distributed Control Systems

Cyber-physical systems are intelligent, adaptive and dynamic systems, organized in multi-agent networks in order to interact and reach a common objective: the multi-agent system goal (also called desired status). 

Monostori in [20] named a cyber-physical system for manufacturing applications a cyber-physical production system (CPPS), i.e., a set of intelligent, autonomous, collaborative and interconnected subsystems, characterized by intelligence, reconfigurability and responsiveness. The field in automation control and distributed artificial intelligence that studies autonomous and reconfigurable industrial systems is called industrial agent control theory. Wooldridge defines an intelligent software agent as “*a computer system that is capable of independent (autonomous) action on behalf of its user or owner*” [21]. 

The traditional approach in industrial control systems is represented by centralized architectures, where a central control node is capable of orchestrating the information flow coming from field devices and dispatching control signals to automation actuation systems having full visibility on the overall process and setup of the system. With the growing complexity of automation and the needs of customization of products, fast response to market requests and reduction of reconfiguration time, this approach is moving to decentralized control systems leveraging on service-oriented architectures, aimed to address flexibility, reconfigurability and responsiveness to realize adaptive control systems [6].

Industrial agents are cyber-physical entities since they represent the integration between physical world perceived via sensors and controlled via actuators and the virtual world represented by the software elaborating sensed data and generating actions on the surrounding environment, leveraging on internal control policies [22].

Industrial IoT becomes in this way an enabling technology in order to create such pervasive and interconnected equipment ecosystem to collect information from the machines and to exchange data among different services, both for field control and monitoring and process optimization purposes; to tackle such complexity multi-agent technology in last years was rediscovered after decades of sidelined [23].

The industrial multi-agent technology is mainly used for exchanging data, elaborated information and services leveraging on service discovery mechanisms and providing high-level signals in relation to online planning and scheduling. These mechanisms for communication in such context was formalized by the Foundation for Intelligent Physical Agents (FIPA), officially accepted by the IEEE as its eleventh standards committee on 8 June 2005 [24]. The system is based on the existence of a common peer to peer platform, defined as a agent platform that has the aim of providing a common bus infrastructure for the deployment, service discovery and message exchange among agents subscribed to the MAS.

### 2.4. Internet of Things and Fog Computing

Industrial IoT systems are a concrete example of multi-agent applications crossing the automation domain (since the field devices have real-time constraints and interfaces with physical hardware) and the pure software domain for data mining (streaming systems and synchronous/asynchronous communication enablers), data cleaning and analytics. In the following section we are going to present two use cases of distributed system application for Industrial smart nodes, both for static nodes (plant air quality monitoring) and dynamic nodes (industrial scooter fleets control) with two different architectural patterns, but in both cases implementing distributed sensor networks, for data retrieving, filtering, analysis, alarm generation.

IoT devices communicate using the TCP/IP protocol, as internet standard. Normally in this application a wireless communication is required, using technologies as Ethernet, Wi-Fi, Bluetooth, ZigBee, radio frequency identification (RFID), or barcodes. Moreover, every IoT device need to have a unique identity; it is possible to use an IP address or a universal unique identifier (UUID), depending on the technology used [25].

Wireless sensor networks (WSNs) are applications with IoT nodes capable of exchanging data using internet protocols both in local network or using internet as infrastructure. It is possible to have a one-way communication protocol, to allow the communication sensor-network, but also a two-way communication protocol [26]. The components present in a WSN are a radio transceiver, a microcontroller, an analog circuit for signal processing, an embedded operating system and a power source. An advantage of this technology is the fact that using these sensors is easy to create an IoT network, because different devices can easily be connected. WSNs are used where there are spatially distributed autonomous sensors, with the objective to monitor the environment and its dynamics in complex system. There are three common way to design a WSN: star, cluster-tree and multi-hop mesh [25] (Figure 2).

Nowadays, with the increase of technological devices, as phones and tablets, there are a wide range of devices connected to the internet available everywhere that make possible mobile computing [27]. These devices have embedded sensors able to collect data, such as GPS location, gyroscope data, acceleration, images, etc., and, a result, they can be integrated with IoT systems [25].

Smart nodes are used in a myriad of fields: buildings, manufacturing, healthcare, transport, bringing the number of devices to more than 50 billion [28]. This development has shown the limits of the cloud, saturating its potential and limiting the most demanding application fields, such as real-time applications. To overcome these limitations, an intermediate layer has been developed, located closer to the physical devices than the cloud and equipped with computational power that enables near real-time control systems, reduces latency, allows the management of many nodes over a wide geographical area. This intermedial layer is called fog computing. The new architecture proposed by the fog computing concept was developed by the OpenFog Consortium. OpenFog is an organization founded by ARM, Cisco, Dell, Intel, Microsoft and the Princeton University Edge Computing Laboratory in 2015 with the aim of overcoming the limitations of the cloud, improving flexibility, security, performance, bandwidth [29]. In this way, requests with stringent timing are managed in the fog, while those with soft timing are managed in the cloud. These characteristics are often located on a gateway, as illustrated in Figure 3, that collects all the data from the network of devices to which it is connected, analyzes them and sends them to the cloud [30].

## 3. Architectural Patterns for Smart Nodes Design and Orchestration

### 3.1. Common Architectures and Software Enablers for Industrial IoT Applications

With this article we would like to present a general-purpose framework to implement an IoT sensor network application in order to answer how to implement Industry 4.0 distributed systems paradigm. General ideas driving the research are the fact that SMEs generally struggle with the complexity of technologies of Industry 4.0 and their costs. For these reasons we focus on applications characterized by open hardware and open-source software without hard real-time constraints, that we will analyse in dedicated future research.

Distributed software technologies are central since they provide technical tools for the implementation of distributed and multi-agent industrial applications that nowadays leverages especially on distributed service architectures typical of internet development domain (e.g., REST) [31].

Today software engineering for distributed systems is focusing on microservices technologies, leveraging on peer-to-peer architectures based on broker-based systems bus that are very well coherent with the mechanisms of agent communication described by FIPA.

In our research we are focusing on using common patterns both for distributed automation (field level) and software distributed services, scaling the same approach on different levels of the IoT ecosystem. According to articles and specialized websites on embedded systems based on microcontrollers for IoT applications, C language remains the most used programming language to develop the software for platforms hosting sensors and Wi-Fi modules. Thanks to the rapid spread of single-board computer for low power applications (e.g., Raspberry PI, Banana Pi, etc.) high-level and object-oriented languages are more and more applied in this field like Go, JavaScript, Python and LUA [32,33]. Using HTTP based protocols helps in designing platform independent solutions and we leverage on this to interconnect low-level (data sampling and actuation) and high-level services (business logic, analytics detection).

For industrial services and software microservices, following the modern approach to develop server applications in a decentralized and scalable way, two different communication approaches are used in order to implement SOA architectures: synchronous and asynchronous protocols enabling respectively the request-response and the event-driven patterns [34]:(1)Synchronous communication is a blocking protocol, i.e., the client is blocked until it receives a response from the server application (Figure 4). Modern distributed applications leverage on the RESTful approach, where APIs are used for inter-platform communication with, generally, JSON data exchange, leveraging on top of HTTP internet protocol [35,36].This is a pattern we decided to use both to manage data exchange among different functionalities internally to the web applications (developed to implement the HMI and visualization systems), but also to create the IoT platform, connecting the field device to the remote server for data transferring, storing, alarm notification.(2)Asynchronous communication is a non-blocking protocol and it enables the event-driven pattern. The relationship between producer service and consumer service can be one-to-one or one-to-many. Generally, in multi-service applications, the multiple receivers pattern uses architectural styles based on publisher-subscriber and event-bus systems (Figure 5). Publisher-subscriber architecture is based on asynchronous communication of services inter-platform. The orchestration of information exchange with an event-bus orchestrator is also defined choreography pattern. We decided to use such pattern in the first use case to manage the different services in the internal software architecture implemented both in the sensor node tracking the device and the datacenter system and in the second use case for the orchestration among the different layers of IoT infrastructure (1st layer: edge-fog, 2nd layer: fog-cloud/data center). In comparison with other publish and subscribe protocols, Message Queue Telemetry Transport (MQTT) is a more lightweight broker-based protocol, a good balance between simplicity, great level of abstraction and cyber security; MQTT is also an ISO standard protocol (ISO/IEC PRF 20922) [37]. This protocol is applied in a wide variety of industries: automotive, logistics, manufacturing, smart home, consumer products, transportation [38]. The key concept of this message queuing is the message topic that represents the identifier of a message entity. Another important aspect of this publish/subscribe broker-based mechanism is that a software application can represent at the same time a data provider and a data consumer depending on the topic: this exceeds and strengthens the typical client/server communication allowing a bidirectional communication, that will be more and more important for distributed control signal exchange that is expected in near future thanks to high bandwidth and low latency of 5G [39].This choice comes from the lightweight and easy implementation of publisher consumer nodes with broker-based patterns with not stringent real-time constraints both in terms of latency and bandwidth. Our goal for future research is to scale such approach also to more demanding manufacturing applications, implementing DDS [40] or RTPS systems, but maintaining the same functional architecture. In this way we will be able to manage also configurations requiring continuity of the services against failures, thousands of messages per device per second [41].

### 3.2. Web-Based Dashboarding for Data Visualization and Analytics

In the proposed architectural pattern, we include also HMI systems in order to provide graphs, alarm interpretation and basic analytics to the users in order to interpret the complex distributed system status. Since the overall pattern relies on internet protocols, we decided to implement web-based frontend applications.

To follow the service pattern also in the web app, a clear distinction between front-end services and back-end services is applied. To test the web application usability, we asked to low computer-skilled people to test them, in order to understand and assess the usability of the interfaces [42]. Internally to assess the technical implementation usability and functional tests were performed using different browsers, as Google Chrome, Apple Safari, Microsoft Edge, Mozilla Firefox, according to the analysis of [43].

## 4. Industrial Use Cases for the Application of Smart Node Sensor Orchestration

In this section we aim to present two different use cases developed by our research group in the context of a cooperation between the university of engineering and the local chamber of commerce finalized to the support of small-medium enterprises with difficulties to introduce I4.0 tools, due to lack of competences and ideas of possible improvements in their area of business. The local chamber of commerce scouted 17 different enterprises interested in this pilot and exploration; together with the department of engineering the interview phases started in order to discover main (and possibly hidden) needs to be addressed with new technologies. The needs collected with the interviews were clustered as follows: 12 companies interested in IoT to enable data analytics and data exploration, three in collaborative robotics, two in deep learning and computer vision applications. 

It is interesting to highlight the fact that the companies requiring IoT and data analytics were not only traditional manufacturing industries, but also companies operating in the health and monitoring services for SMEs or in dairy processes, viticulture, brewery and in general food area. This heterogeneity is a confirmation of what Díaz et al. describe in [44]. To reinforce the knowledge transfer to the selected companies, we decided to implement the Agile SCRUM framework in order to keep the personnel of the SMEs in the loop as main stakeholders, starting every sprint with the requirements analysis and refinement phase and inviting them during each system review to collect feedback both on the technological and methodological understanding. The approach was successful since after some sprints of knowledge sharing, the personnel started understanding the potential of IoT and in general I4.0 tools and figuring out new possible applications in their business domain [45].

In general, all these enterprises were not interested in hard real time data streaming technologies due to the fact that they were interested to use these data analytics to better understand possible patterns (sometimes known for experience but not confirmed by scientific analysis) in order to reinforce human learning driven by data, as explained in [16], and support their decision process, without full automation and automatic control closing the loop. Additionally, they were interested in cheap solutions achievable via wireless devices in order to use IoT nodes as plug and play systems to improve the flexibility without adding wirings in their operative environments.

Two selected use cases from the first cluster are presented as proof of concept (POC) in order to demonstrate the end-to-end design flow; we used this general workflow starting from requirements analysis, functional breakdown, functional architecture design, and detailed design with a physical architecture. We present these two different use cases in order to cover the scalability of the methodology, focusing on applications without critical real-time requirements (that we will cover in future research). Open software and platforms are used (Raspberry Pi, Arduino libraries, etc.) in order to lower the costs, with a wireless communication in order to have plug and play systems easily installable.

### 4.1. Use Case 1: Industrial Vehicle IoT Monitoring Platform

#### 4.1.1. Stakeholder Requirements and Business Hypothesis

The target of this use case is to create a condition monitoring central platform that can generate events about the status of a vehicles fleet. A fleet of vehicles is difficult to control and maintain, the vehicles are always on the move, distributed in places not always close. Monitoring all vehicles in use from a single platform is useful, as it allows one to understand how they are used, to perform predictive maintenance based on the data collected in the field and to know their position in real-time.

In this use case, every vehicle in a fleet was endowed with an IoT device equipped with computational power, with the purpose of acquiring data, processing and sending them to a central server for data saving and visualization. The configuration of the acquisition systems must also be centralized, to make the system flexible and to avoid repetition of tasks considering vehicles of the same type.

As mentioned above, one of the main consequences of this use case is the capability to facilitate maintenance, in order to avoid downtimes due to breakage of components. This application represents an effective tool for both preventive and predictive maintenance since it provides a decision support system both with visual representation of trends and the capability to manage alarm thresholds. Preventive maintenance is based on statistical indices, exceeded which a component is likely to break (e.g., maximum number of kilometres, number of braking, maximum temperature, etc.). For each vehicle, you need to set thresholds and be warned when a measurement exceeds this value. Predictive maintenance, on the other hand, is based on the analysis of data collected by vehicles and applying recognition of anomalies.

Retrieving data coming from the motor controller becomes the first enabler to create such distributed infrastructure. A motor controller is a device with the objective to coordinate the entire vehicle system and to improve the performance of an electric motor.

The controller manages all the information about the vehicle and its motor, such as the vehicle speed, the engine temperature, the motor current, and others. Those data can be collected using the controller area network bus (CAN-bus), that is a vehicle bus standard; it was designed to allow the communication between microcontrollers and devices without use a host computer. Other data can be derived aggregating the collected data; an example could be the electric power absorbed by the motor that could be calculated from the current and the voltage of the engine. 

With this system a company can have a clear overview of its vehicle fleet and its use. Moreover, with this amount of collected data, if a pattern between some metrics exists, it will be found by the data analysts and it will be useful to have a prediction of the vehicle’s status. Another important aspect is that knowing how the machine is used will help the designers to optimize what kind of motor and controller is the most suitable for every kind of vehicle, representing a feedback for continuous learning.

All the data collected are available in a web-based dashboard that can show the real-time series or the historical data. Through the website it is possible to manage all the vehicles of a fleet, set alarms and notifications or add new metrics to be collected.

#### 4.1.2. Functional Architecture

The functional architecture (Figure 6) shows IoT nodes, that collect data from the controller via CANopen protocol, send them to the server after a prefiltering process; the server stores the data and the alerts, and it shows them in dedicated dashboards exposed via website. The user can see all the collected data (both historical data and real-time stream coming from each field node) and alarms from the server, choosing the vehicle to analyse. Otherwise, it is possible to see the real-time data of a vehicle by connecting directly to the node device using its Wi-Fi hotspot.

In this use case a platform that allows the user to set all the vehicle parameters was designed in order to ease the monitoring of the entire system. Alerts, notifications, data to collect, vehicle information are set by the user on the server web page. We predefined variables that the user can choose, but it is possible to set all the variable available in the CAN network. The user has the possibility to set alarms and notifications. To do that he chooses a variable and a threshold; every time that the chosen variable exceeds the threshold, an alarm is generated on the vehicle and sent to server. If the user wants to receive an email notification when the alarm is active, he will set a notification on the alarm directly on the server.

#### 4.1.3. System Architecture and Detailed Design

In this project the system architecture (Figure 7) presents two different physical subsystem to consider:(1)The physical device installed on the machine collects the data since it is connected physically with the vehicle. The onboard device is a single-board computer (in this use case a Raspberry Pi is used) connected via CANbus to the vehicle-bus to which the controller is connected.(2)The main server could be hosted in an on-premises data center or in a cloud virtual machine. In this server there is an HTTPS Server that manages all the requests coming from the fleet and the web server, and an MQTT broker to which the notification and the web server are connected to handle, respectively, the notification sending and the real-time visualization. Using the website, the user can manage and monitor the vehicles fleet, configure the alerts, set the acquisition and the algorithms; this information is reported in a configuration file that a vehicle must download.

##### The Physical Device

We can divide the device into different modules, as represented in Figure 7.

Acquisition moduleThe CAN protocol has three different levels: physical layer, transfer layer and application layer. It does not have a standard application layer, and in this use case the CANopen protocol is used as application layer. This protocol has the possibility to have a standardized embedded network; very important is the fact that this network has a high capability of flexible configuration [46]. The data to collect from the CANbus are available in the configuration file for the acquisition module. Data are published with an MQTT publisher, that is used for the internal comunication from the different modules. Prefiltering moduleCollected data are read from the prefilter module that discards the data that don’t change and updates only the timestamp with the time of the last collected value. All the data are collect to form a JSON object. Every 30 seconds the JSON object is written on a FIFO queue that links the prefilter module to the sender module.Algorithm moduleThe algorithm module, exploiting the configuration file coming from the server, reads only the interested data using an MQTT subscriber. The configuration file contains all the algorithms that the user configures from the server, and it is downloaded from the vehicle everytime it turns on. The outputs of the algorithm module are published with an MQTT publisher, and they are read by the prefilter module like any other collected data from the vehicle. In this module standard algorithms are implemented both to implement algebraic operations to combine signals, but also standard analytics (mean, standard deviation).Alert moduleUsing the same paradigm of the algorihm module, the alert module reads the interested data knowing them from the alarm configuration file coming from the server and set by the user. An alarm occures both when it starts and when it ends. The alarms are values that exeeds a threshold. When an alarm is detected, it is written as fast as possible in the alarm queue.Sender moduleThe sender module reads periodically the two FIFO queues and sends the queue content to the server as JSON via REST API call. The internet connection is available thanks to the 4G modem installed on the Raspberry Pi. If the internet connection is not available, data and alerts are saved in a SQLite database (acting like a temporary buffer), and when the connection returns available, everything that is present in the database is sent to the server.Website moduleThe user can access to real-time data directly from the vehicle web page (Figure 8), connecting to the local Wi-Fi network hosted by it, or using the web page exposed by the server, where the user can see the data of all the vehicles. The website has a MQTT subscriber from which it reads the data and the alarms published.

##### The Server

The main server hosts different processes to fulfill all the requirements of the project. These software components are:The HTTPS server that has two main goals: enable the communication with the vehicles fleet and manage the requests of the website. This module communicates via REST API. The server can distinguish between different type of message exploiting the header of the API request, e.g., headers: storeData and storeAlert are used by the vehicles to send data or alarms. When a new request is received, the server has three possible actions. The first one is the data saving. The received information is saved in a relational database (MySQL) using the CRUD module. The second one is the data sending. Every time a vehicle turns on, it requests its own configuration from the server. So, the HTTPS server, via CRUD module, sends back the right information. The third action is forwarding the message of data and alarms to the MQTT Broker, allowing the website subscriber to show them in the dashboard (Figure 9).A CRUD module is used to manage the MySQL operations. Each topic acquired by a vehicle has its own table, automatically created. In this way it is easier to query the data for the visualization. Also, all the information regarding the alerts, the acquisitions and fleet metadata are saved in MySQL.If an alert is connected to a notification, it is sent via an MQTT publisher. Through the MQTT broker the notification is forwarded to the notification server, that warns the user with an automatic email. Using the website, it is possible to configure new alerts, set thresholds and create notifications linked to a specific alert. It is also possible to see the alert history and the active alerts, so alarms that are active in that moment (Figure 10).The website module is very important in this architecture because it links the user to the system, and it allows the user to manage each element of the architecture. The design phase was realized considering the 16 principles for noise eliminations explained in [47]. The definition of noise in the context of UX design is everything preventing the right message understanding: in general, the principles address good legibility, privilege simple structure both of the website framework and of the text. The website designed for the use case allows an easy management on the vehicle configuration; data, alerts, algorithms, and notifications are managed from the respective web pages. The user can add, modify or remove these features, and the vehicle will download them when it turns on. A user can see the real-time data published from the vehicle thanks to an MQTT broker. He can see in graphs the past data collected, because the website interacts with the HTTPS module that queries the database via the CRUD module.

To link the physical vehicle with the server, the user has to create a virtual vehicle on the server obtaining a unique vehicle ID; the vehicle ID is required when the user installs the device for the first time. Every single information of the vehicle is saved on the database with the vehicle ID information.

To facilitate the management of the system, it is possible for the user to create vehicle groups in the server that allows to set alarms, notifications, algorithm to all the vehicles of the chosen group. This is implemented to avoid the user having to create more times the same features for every node. The structure of the website map (really similar to the organization of the website realized also for the second use case) is reported in Figure 11.

For both the website on the server and the vehicle, we used Node.js for the back-end, while for the front-end, we used HTML, JavaScript and Bootstrap, a CSS framework. The communication between the front-end and the back-end is implemented using AJAX requests, allowing asynchronous update of the pages.

#### 4.1.4. Verification and Validation

In order to verify and validate the application we tested the integration between different modules and then the entire system data flow, applying in this Wireless sensor network the testing pyramid methodology [48] to reduce the risks of software bugs, and incapsulating the interaction with physical sensors in libraries that, once debugged, are not changed.

We create fake data after testing the physical acquisition via CANbus. In the future a new development could include some sensors to collect new data; these sensors, as accelerometer, GPS, gyroscope, magnetometer, etc., could increase the data to collect and the information about the vehicle. With this architecture it is easy to link some sensors to the Raspberry Pi. After the technical verification, we verified some qualitative KPIs:Simplicity in the configuration process; in this POC one of the key elements is the flexibility of the system, that allows to define the data to collect, the algorithm to calculate, the alert to trigger and the notification to send, directly from the server. To test this part, we asked to three non-expert users (done by three alpha testers not involved in the prototype development), to use the configuration page in the server, explaining only the feature of the website but without providing advices.Availability of the system on a week time range; during a real test monitoring an industrial vehicle available in in the ArsControl Laboratory (UNIMORE) no downtimes of the application was identified having a 24 h availability of the monitoring system. The vehicle used for the test is an industrial scooter.Simplicity in the website dashboard understanding; we asked to the same three non-expert users to watch the two dashboards, one onboard the vehicle and one on the server, and to interact with these; this experiment shows that the dashboard was clear and intuitive.

### 4.2. Use Case 2: Plant Air Quality Monitoring via Wireless Sensor Network (WSN)

#### 4.2.1. Stakeholder Requirements and Business Hypothesis

With this second use case we decided to improve our previous architecture in order to be able to have a system that can generate feedback signals for IoT field nodes to act on detected issues: to reach this goal the signals regarding alerts and alarms are not directly handled by automatic alert systems as in the previous use case, but are published in the System Bus that we are going to discuss in details in order to be consumed by field nodes subscribed to manage these specific events. 

One of the pillars of Industry 4.0 and cyber-physical systems is how to keep human operators in the loop, creating a collaborative environment to support the operations of human beings. Supporting practical activities and tasks is not the only way to reach this goal: also the maintenance and guarantee of the psychophysical well-being of the staff is certainly one way in which an intelligent system can support human activities. For this reason, this use case aims to create a deep and distributed sensorization of industrial plants in order to monitor indoor air quality parameters that are relevant for the health status of personnel working in the shopfloor.

A collaborative system in this field is not only capable of collecting data for visualization and reporting in dashboards but is also capable of identifying in real-time anomalies in the environment parameters in order to alert human operators or to create triggers for other intelligent system delegated to recover the target and desired status.

The scope of this use case was the creation of a scalable distributed network of smart sensors interfaced with room gateways, capable to group and store semantically the signal data of sensors present in the same area and to communicate such data (filtered and with analytics applied) to a central server located in cloud or on-prem with a building information modelling (BIM) representation of the plant, in order to centralize and visualize the full status of the system both in terms of sensor localization and data transmitted.

The BIM basically is a CAD 3d with an object-oriented approach, where it is possible to model both the visual representation of an object but also connect a dataset not necessarily related to infrastructural aspects. In these terms BIM have the characteristic to represent the digital twin of an industrial plant if it is enriched with live data coming from distributed nodes or equipment and it is an effective tool for visualization and root-cause detection. Web-based dashboards for the real-time series and simple analytics were also developed, easily reachable with QR codes encoding the URL of the local website for the room exposed by the web server hosted on the hub gateway.

#### 4.2.2. Functional Architecture

The functional architecture (Figure 12) is composed by different IoT nodes, capable of collecting data from the field via physical sensors, sending it to the room gateway that acts like a fog node to analyse, store and displaying it in the form of visual dashboards describing a website. The IoT nodes collect data on trigger via a request of the room gateway, hosting their databases.

The fog nodes are capable of redirecting the information collected and filtered to the main server where the BIM model is stored. This server can be hosted on-prem or on a cloud environment. Only the last updated values are stored in the BIM server for each sensor in order not to overload the model, since it represents the single possible failure point of the architecture. In the BIM model is also stored the URL to reach the website exposed by the web server hosted on the room gateway to consume and visualize also historical information.

The BIM model, as new data are sent by the fog nodes is updated with a software that interfaces with the room gateways. In the room gateway configuration file, it is possible to define the rules and the thresholds of acceptable range for each air quality parameter: the mechanism implemented is capable of triggering an alarm to send an automatic alert to human operators in charge of repairing and restoring the standard air quality and eventually usable from automatic actuators capable of acting on the environment to improve the status.

#### 4.2.3. System Architecture and Detailed Design

The system architecture (Figure 13) is mainly composed of three different physical subsystems:(1)*The IoT field node*, the edge device in charge to interface with the environment in order to collect data about air quality parameters (in our proof-of-concept temperature [°C], humidity [%], pressure [hPa], VOC [ppb], eCO_2_ [ppm], H_2_, EtOH) or to notify and act on environment when an anomaly is detected by the room gateway with an alert signal notified on the room system bus.(2)*The room gateway*, a fog node with the responsibility of collecting data from the field node, filtering the data, storing the data on a SQL database (in our application we used a MySQL database). In the filtering module there is also a service that, given the thresholds of the alarms inside the configuration file, is capable to generate an alarm on the system bus handled by specific services (email sender service deployed on the fog node and alarm handler on the IoT field node).(3)*The BIM updater service* hosted or in an on-premises data center or in a cloud virtual machine. In this server there is another MQTT system bus where each room gateway is subscribed and publishes new notices.

The entire architecture relies on a common ontology designed to associate the air quality signals to the right source to be referenced in the BIM model: this ontology is specified by the room identifier, the parameter of interest and a number (in case in the same room we have multiple nodes to perceive some measurements). 

This ontology based on the topics exposed on the MQTT broker guarantees a flexible system and an easy scalability of the number of field devices, since there are no direct client-server connections among the nodes of the system.

On each gateway there are different topics exposed both to create data requests and to read new samples available from the field devices publishing parameters of interest, but there is no direct knowledge about the sensor that will provide the information. This simplify the commissioning of the system, since it is sufficient to specify in the configuration file of each node what are the parameters it is going to provide to the infrastructure (derived from the BIM model) and what are the Wi-Fi access point and the MQTT server info and credentials to establish the communication. 

This paradigm, really oriented towards the multi-agent service discovery mechanisms, also guarantees the decoupling of the system on software level increasing the overall reliability and the blocking of the propagation of anomalies: the failure of a field node only causes the loss of data without downtimes of the system.

In the following sections a clear description of the software structure of the three components is provided:

##### IoT Field Node Software Architecture

An embedded platform to host the air quality sensors and the IoT field node software architecture is structured with the following sections:Configuration file with:
Wi-Fi access point info (WLAN_SSID, WLAN_PASSWORD)MQTT Server info (MQTT_SERVER, MQTT_PORT, MQTT_USERNAME, MQTT_PASSWORD)

Method for Wi-Fi client and MQTT client with topic subscription for publishing (Gateway requests) and consuming (Gateway data sending).
Setup method with:
Connection to Wi-Fi access pointLoop repeated until successful connection.Setup of MQTT topic subscription (parameter requests and alarm request)Pin mode definition and Setup method provided as libraries by the producer of the sensors.

##### Loop Method

MQTT_connect method call, in order to establish and maintain the connectivity to the System Bus on the Fog gateway. In case of issues, it automatically tries a new connection in order to reestablish the distributed network. In this method it is possible to specify the maximum number of times to try a new connection with the broker and the time distance between the different attempts.State machine to manage the different requests received by the room gateway: since in this application there are no hard real-time constraints due to the slow dynamics of the parameters, we decided not to implement a temporized loop inside the field device, but rather to start the request on the fog device that can be eventually changed in the configuration file accessible and modifiable via the web-server. Leveraging the topic request ontology, the sensor understands what sample is required by the gateway and calls a specific method to retrieve a new measurement of the parameter specified in the topic (since the nodes generally have multiple sensors connected to the input pins of the microcontroller board).

In our proof-of-concept we used as embedded system hardware a NodeMCU board with a ESP8266 microcontroller, a BME680 sensor by Bosch (IAQ, temperature, humidity, pressure) and a SPG30 by Sensirion (VOC, eCO_2_, H_2_, EtOH).

##### IoT Room Gateway Software Architecture

The room gateway is implemented using one single-board computer (in our application a Raspberry Pi 4) hosting the MQTT Server (Apache Mosquitto), the Apache webserver hosting the dashboarding system, the MySQL database server for the fog node and finally the services and software modules described in the following:Configuration file: defining connection string and authorization credentials to the MySQL database for the field devices, MQTT server info, Wi-Fi access point information and credentials, air quality parameter thresholds for sensitivity and acceptable ranges (applicable for all the devices exposing the specific measure), sender and receiver email addresses for the automatic notifications in case of anomalies.Setup method to define the callback functions that are used when MQTT broker events happen (e.g., connection to the system bus server or data published on a subscribed topic of the broker).MQTT message handler capable of:
○Classifying the source of information leveraging on the topic analysis with the explained ontology in place [room identifier; air quality parameter; source number]. Not recognized messages arriving on the System Bus topics and violating the ontology are filtered out and not handled.○Calling an air quality parameter analyzer (data storage function).Class to manage CRUD operations on the MySQL database, in order to dynamically create during the initialization a database on the server with the identifier of the room specified in the configuration file, to generate specific tables only if a first message about an MQTT topic is received (if topics for a certain topic present on the system bus and following the ontology are not used missing the field nodes, tables are not created).Methods for data storage, embedding a business logic to consider the total amount of data of the circular buffer implemented on the fog node, the decision between new insertion in DB (delta in two consecutive sample values greater than the sensitivity threshold in configuration file) or update of the last sample timestamp in DB (delta in two consecutive sample values lower than the sensitivity threshold in configuration file).Alarm notifier module to detect a rising edge in terms of violation of acceptable value range of the air quality parameter of interest (information stored in the configuration file and associated to the sample via the topic-based ontology). Only rising edge outside a “silent time period” generated for a specific air quality monitoring after the rising edge detection is capable to generate an alarm trigger.Email sender service triggered by the alarm notifier to send an email to addresses specified in the configuration file of the module to notify which room and which parameter was outside the acceptable range and reporting also the detected deviation and the timestamp.

All these functionalities are called by the main application that basically in the initialization creates the MQTT connection both as data publisher (request generation for each parameter represented by a topic of the system bus) and data consumer (to read data published by the field devices subscribed to the MQTT broker) and the parallel threads (with frequency defined in the configuration) to manage the measures to be handled. To create the MQTT client the Apache Paho library was used. Then a loop, so that the scheduling task keeps on running all time. 

The exit condition from the loop status can be represented by:(1)Scheduled task in pending to run (that will trigger data requests for air quality parameters)(2)New message event notified by the broker and to be handled as explained before.

Also in this use case we implemented a website hosted in an Apache web server installed on each room gateway. The website offers the possibility to choose the air quality parameter to be analyzed, both with an historical perspective or with a real-time update mechanism [Figure 14], based on asynchronous AJAX callbacks between client and server.

The URL of the website is reachable via a QR code printed on the field nodes present in each room (in this case pre-selecting the parameter in the main page) or via the BIM visualizer: one parameter of the model is the web-site URL. In the graph, shown after the selection, the time series of the parameter of interest and the statistical values are reported (and updated in the case where real-time analysis is selected).

##### BIM Updater Service Software Architecture

The BIM updater is a software agent hosted on the BIM server: similarly to the software services present on the room gateways, it is capable of subscribing to the topics related to the different rooms (exposed via the room i-th MQTT publisher hosted on each fog node). The scope of this service is very simple: to update the IFC document (representing the dataset of the BIM model, with object structure [ISO 16739-1: 2018]) present on the BIM server as new data are available from room gateways. 

The BIM updater identifies the parameters of interest of each BIM model components and updates the IFC file, storing the updated value, the mean and standard deviation on last N samples (defined in the configuration file of the room gateway) and the URL to access the room gateway website.

In the IFC file the properties related to a BIM objects are represented with “IFC property single value” parameters. To guarantee the right association between topics exposed on the system BUS for the BIM updater and the object present in the BIM model, we used the same ontology in the definition of IFC property single values (Figure 15):Topic ontology:Room_number/Parameter/number (number used to distinguish sensor nodes in the same room sampling the same parameter).BIM IFC ontology:IFCPROPERTYSINGLEVALUE (‘{Room_number}_{Parameter}_{number}_LASTVALUE’, $, IFCREAL(value), $);

#### 4.2.4. Verification and Validation

To validate the proof of concept, we used the same paradigm that we used in the previous use case. Once debugged the physical interface with the real sensors we realized fake sensors via software data injectors, capable to trigger alarms. We used these fake injectors to close the verification loop: it was possible to compare the trends in Matlab and in the real dashboards and to verify that the expected violating sample was generating the Alarm notices on the MQTT broker.

After the technical verification, we verified some qualitative KPIs:Simplicity to define and apply the configuration parameters of new fog node and new field node with sensors. With the current POC a manual change of configuration file is necessary, but it is an easy task since there are no configuration parameters hardcoded in the source code. In future development we would like to create an interface between web application and configuration file in the fog node, with the possibility to change also field node configurations via a dedicated topic on the MQTT broker.Availability of the system on a week time range: during a real test of air quality monitoring in the ArsControl Laboratory (UNIMORE) no downtimes of the application was identified having a 24h availability of the monitoring system.Robustness of the system against field nodes failures: we tested the robustness of the architecture against possible failure of field nodes. This test proved that a local failure does not propagate thanks to the indirect data retrieval via service-oriented exchange. In future research we would like to investigate a multi-agent reorganization of room gateway assuming possible power failure or downtimes in the fog node, with temporary storage of external field nodes and a subsequent data transferring from temporary gateway to main room gateway.Simplicity of the website for non-expert users (done by three alpha testers not involved in the prototype development). To test the user experience, we performed alpha test session with three different users, explaining only the feature of the website but without providing advices. The experiment highlighted the high intuitiveness of the dashboards and of the website map, since each user was independent navigating on the different web pages available.

## 5. Future Research

The term 5th Generation (5G) denotes the set of mobile and cellular phone technologies; this standard defines the fifth generation of mobile phones and compared to 4G/IMT-Advanced technology there is a significant evolution.

In this new network, all the 5G devices that are present in a cell transmit using the radio signal to the local antenna. Mobile devices moving from one cell to another are automatically and transparently taken over by the new cell without loss of connection. Only devices that were specially designed for 5G can use this network; on the contrary, 5G devices can use 4G LTE network. 5G enables a wireless technology with a wide bandwidth rate, more speed, capacity, and less cost per bit than the previous one. The network will have multiple access according IoT architecture.

For a hard-real time IoT implementation massive connectivity is requested, with complete coverage, high reliability and low latency. Even high privacy and a high level of security is required. 5G technology will increase the data-rates with a better coverage than the previous one, making possible those new business models. [49]

Leveraging on recent papers on the theme [50], we expect that IoT networks will be adopted more and more and, for this reason, it is important to propose and test standard procedure to design, control, manage the lifecycle and the configuration management of such applications with a systems engineering methodology.

For this reason, our future research will focus on a standard approach starting from requirements management to lifecycle management of complex distributed systems, in order to define systems engineering guidelines. In particular we will focus on:(1)Defining what type of automatic control is implementable via 5G technology with a particular attention to equipment cooperation and IoT control signals exchange, in order to cover the area of remote real-time control with the goal to understand what functionalities and control actions needs to be implemented on the edge and which ones can be delocalized on the cloud or on fog nodes.(2)Investigating how to scale the approach to wireless sensor networks not based on internet protocols in order to design with the same approach networks with low power sensor nodes.(3)Studying continuous integration and testing methodology also including field-level devices, since DevOps methodology is more and more used by companies in software development, in particular in pervasive computing systems and SOAs where the boundaries between automation software and web software becomes less rigid and new releases have shorter time to market with high quality requirements in terms of reliable integration.(4)Studying new legibility UI systems for human interpretability and acceptance of distributed AI systems, in particular where multi-agent actuation nodes are present. This fact could really improve the acceptance of AI systems in a social environment, where generally personnel with low skills in modern IT domain has difficulties in trusting systems supposed to substitute the human jobs and where the complexity creates issues in understanding the choices of agents that are, on the contrary, thought to support in cooperative way human daily work.

## 6. Conclusions

Industry 4.0 provides several technologies with potential, but nowadays companies clearly don’t know how to implement them and how these technologies can improve their business. To answer to this gap, this research article as the objective to propose a general purpose and flexible framework to implement an IoT WSN, following multi-agent and cyber-physical system design principles. The output achieved is in the context of a knowledge transfer project realized by the university to introduce I 4.0 technologies in SMEs that struggle in the adoption of such advanced IT solutions. Such knowledge sharing was organized using Agile SCRUM framework in order to have in the loop company personnel as main stakeholders for methodology and technology learning. The presented use cases have the possibility to be scalable and implemented in different applications (to reinforce this idea, we demonstrated how to apply the concepts in two really different industrial use cases).

The first use case describes a central platform for a condition monitoring system with event generation. In the first system a synchronous communication with REST API is used; this is the main limit that we found in this application, because with this technology a bidirectional communication is complex to implement and this does not allow the system to act autonomously. This communication is correct if only a condition monitoring system is required. The second use case differs from the first one because it also has the possibility to provide feedback to each node in order to manage issues and restore the target set point. In the second use case, to allow a bidirectional communication, we use an asynchronous architecture scaling the orchestration model from high-level software components to low-level field nodes.

Finally, verification and validation methodologies and main KPIs, used as success criteria, are described. Such a framework is mainly for addressing IoT solutions based on cheap embedded systems, where there are no critical real-time constraints, limiting the adoption in critical process control applications. This is one of the future research topics that we will cover both with different protocols (e.g., DDS, OPC UA, RTPS) and exploring the potential of 5G. Another topic to address with additional research is the scalability of the approach towards low power devices that generally leverages on different protocols like ZigBee, Bluetooth, etc.

## Figures and Tables

**Figure 1 sensors-21-01624-f001:**
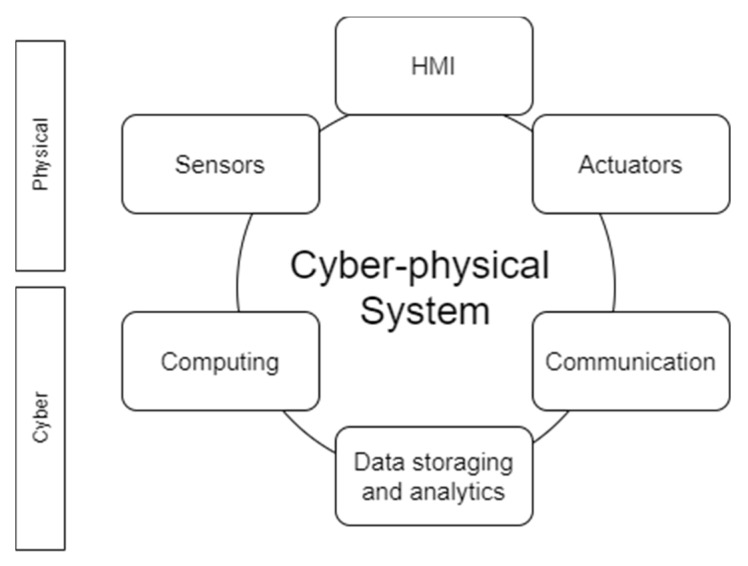
Cyber-Physical system key characteristics.

**Figure 2 sensors-21-01624-f002:**
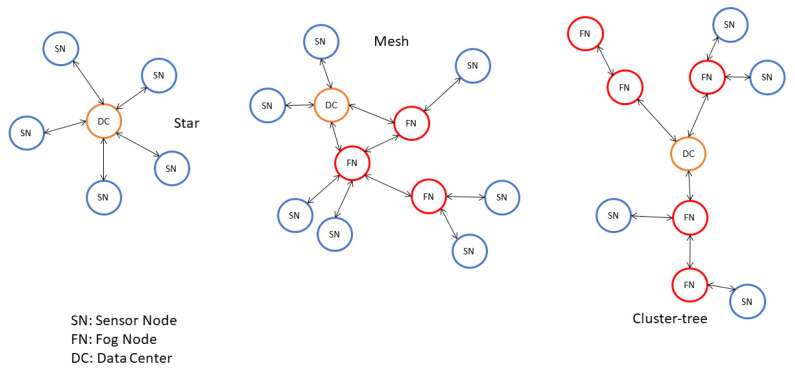
Star, mesh and cluster-tree architecture schema.

**Figure 3 sensors-21-01624-f003:**
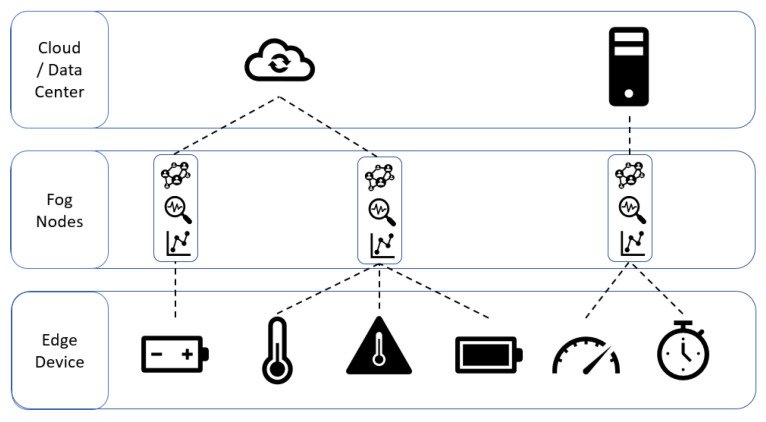
Multi-layered IoT infrastructure.

**Figure 4 sensors-21-01624-f004:**
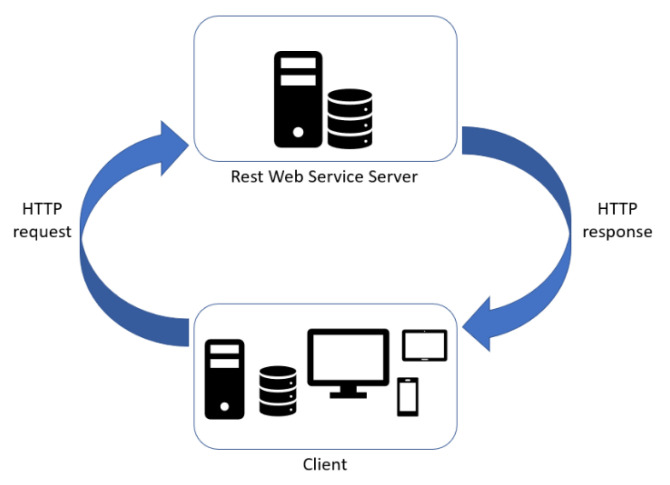
REST API synchronous communication schema.

**Figure 5 sensors-21-01624-f005:**
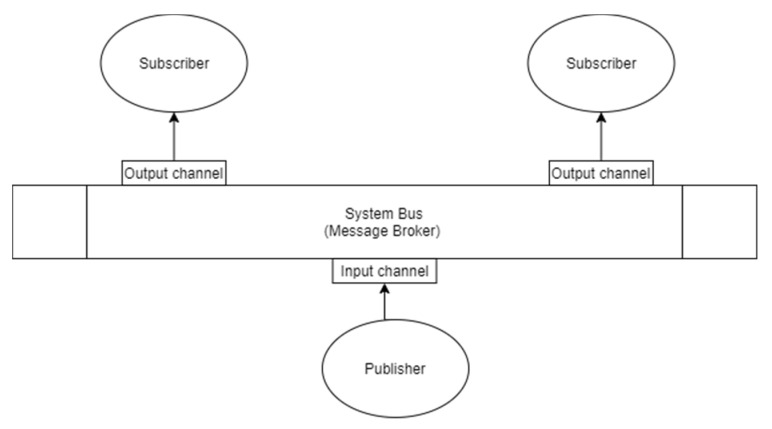
Publisher-subscriber architecture based on asynchronous communication of services inter-platform.

**Figure 6 sensors-21-01624-f006:**
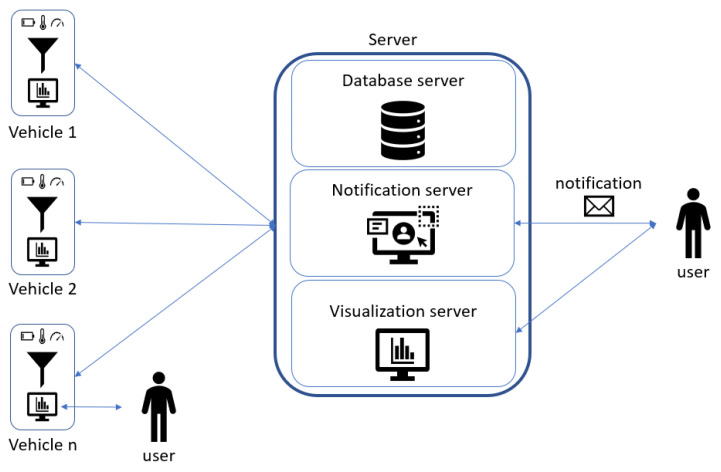
Functional architecture.

**Figure 7 sensors-21-01624-f007:**
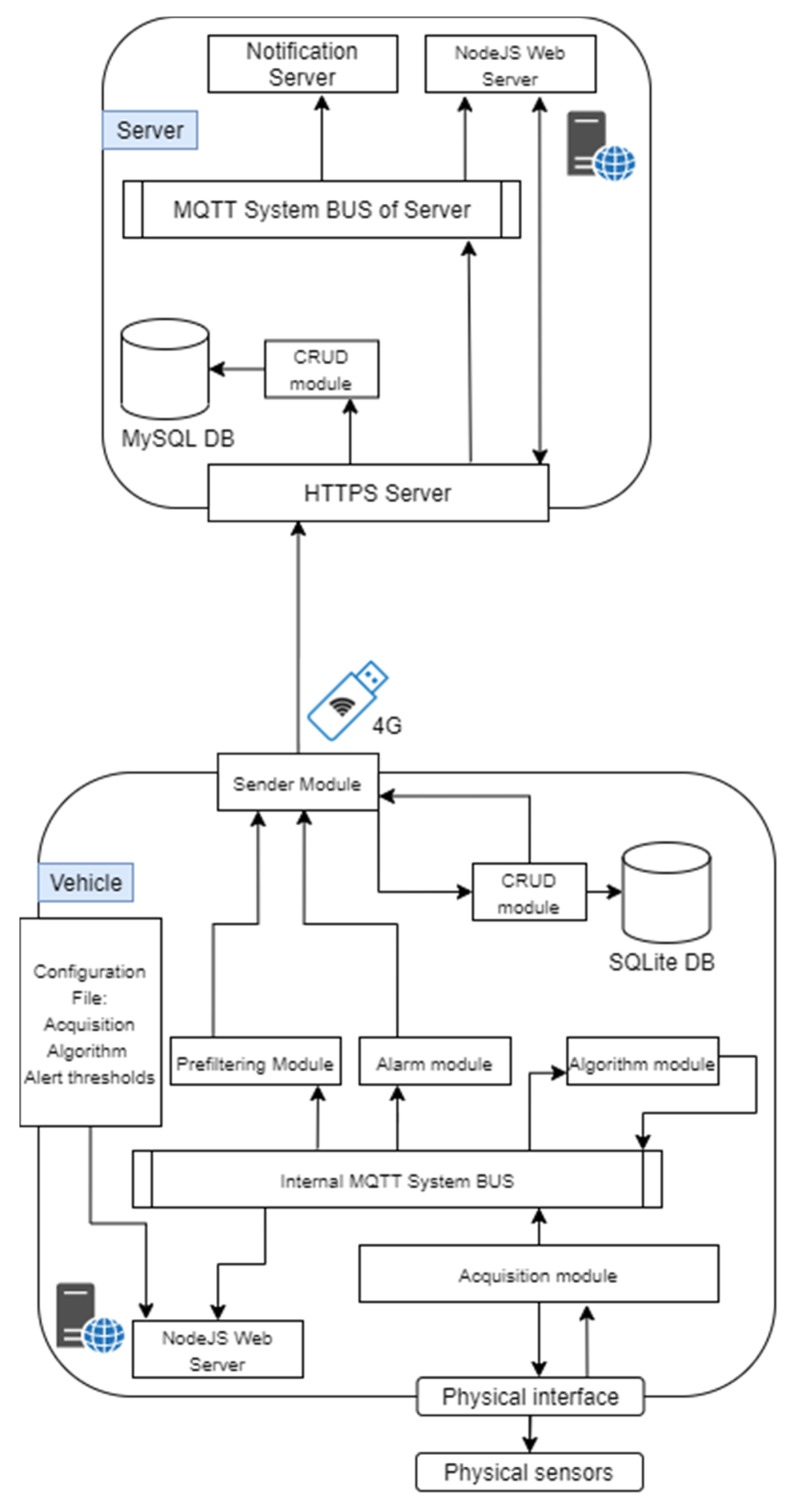
System architecture.

**Figure 8 sensors-21-01624-f008:**
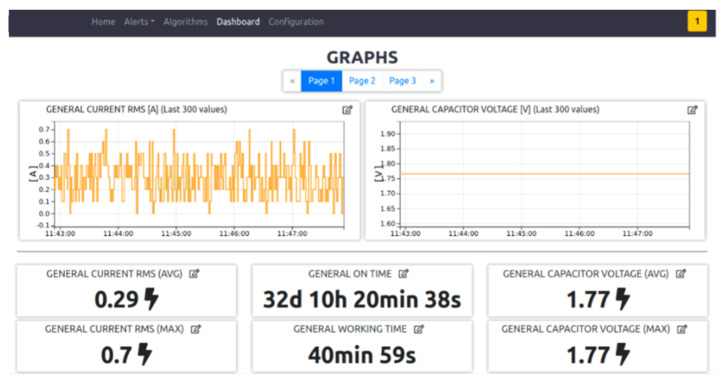
Vehicle dashboard page.

**Figure 9 sensors-21-01624-f009:**
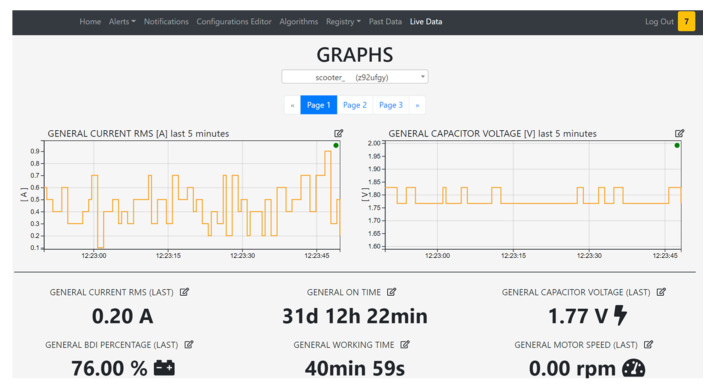
Server dashboard page.

**Figure 10 sensors-21-01624-f010:**
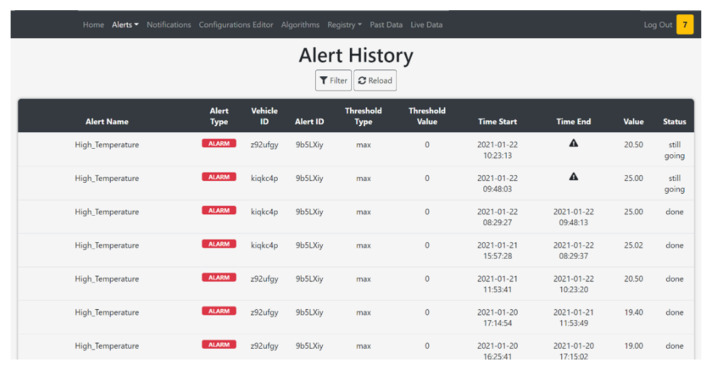
Alert table in server page.

**Figure 11 sensors-21-01624-f011:**
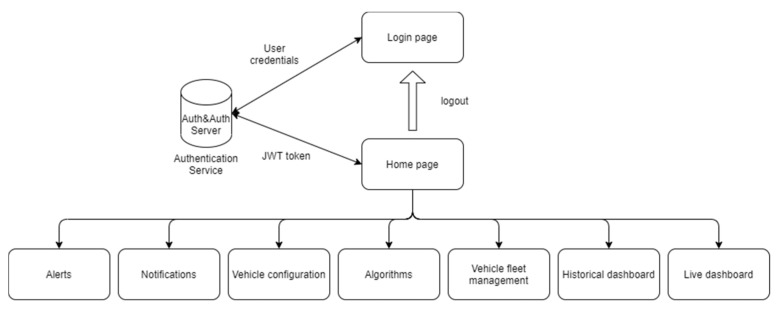
Website map of the HMI application.

**Figure 12 sensors-21-01624-f012:**
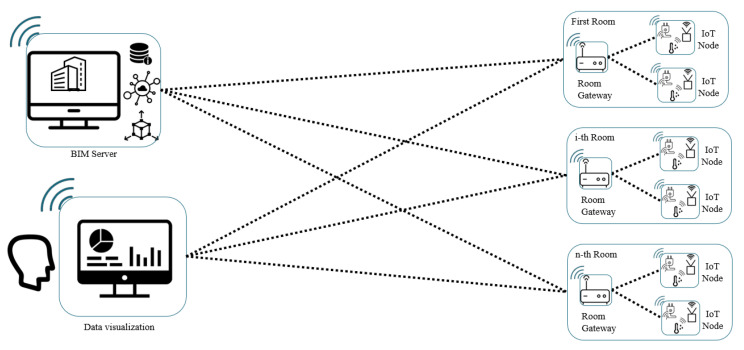
Functional architecture.

**Figure 13 sensors-21-01624-f013:**
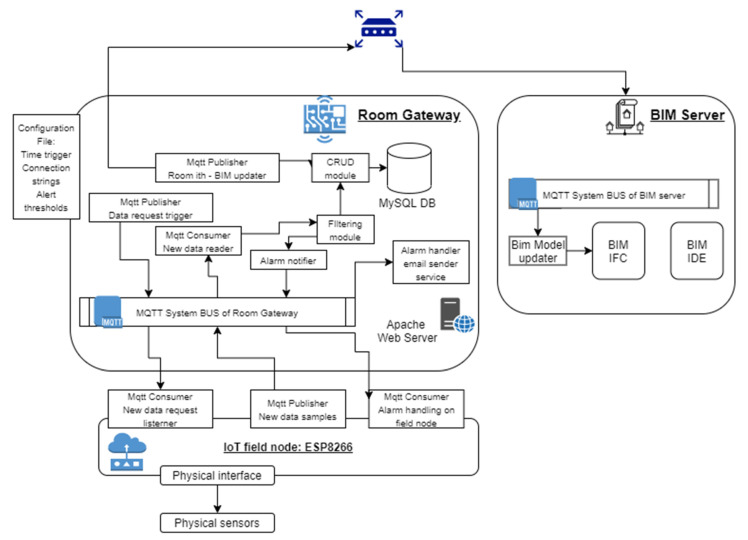
System architecture.

**Figure 14 sensors-21-01624-f014:**
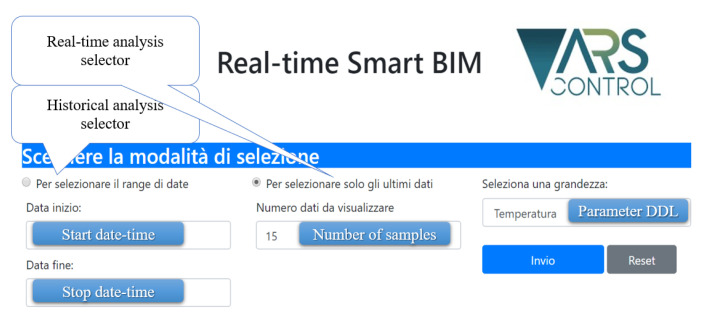
Website main page.

**Figure 15 sensors-21-01624-f015:**
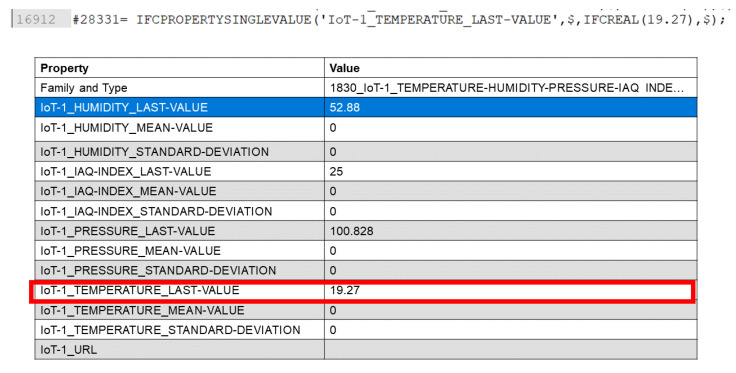
Debugging phase of BIM updater: data retrieved from System BUS, saved into IFC file and then visualized into the BIM model visualizer.

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
