# Peer review of "Smart Node Networks Orchestration: A New E2E Approach for Analysis and Design for Agile 4.0 Implementation"

_sensors, 2021, doi:10.3390/s21051624_

Round 1

Reviewer 1 Report

  • The title is quite vague and might be better specified to what is actually done in the article.
  • The introduction does not derive a clear research gap that this manuscript covers. What is the novelty compared to extant research and why do we need this paper? Further, no references are used.
  • The description of Industry 4.0 in section 2.1 is not complete, not based on enough references, and does not link well to the rest of the paper.
  • It must be better reasoned why this use case and method was chosen, how it is representative and generalizable for other use cases.
  • The following section 4 must be structured better, guiding the reader what is done where. Further, the headings and subheadings used here must be unified, some of them are outside the numbering scheme.
  • A discussion with extant literature and what this paper contributes is almost completely absent.
  • Future research must be better connected to the results, it seems to falls from the sky, while interesting as such.
  • Section 6 is a sum-up, but does not cover contribution, practical implications, or limitations.
  • The list of references is quite low, the article needs much more journal or peer-reviewed references to embed and discuss its results. Further, some references do not have the adequate referencing style (reference 1 is incomplete) and there are a lot of online reports and not peer-reviewed conferences, but too little academic references for an academic article. Please find several suggestions below that should be used to embed the article better in literature:

Industry 4.0 and humans/workers:

Müller, J. M. (2019). Assessing the barriers to Industry 4.0 implementation from a workers’ perspective. IFAC-PapersOnLine, 52(13), 2189-2194.

Rauch, E., Linder, C., & Dallasega, P. (2020). Anthropocentric perspective of production before and within Industry 4.0. Computers & Industrial Engineering, 139, 105644.

Industry 4.0 and value-chain spanning aspects/Interconnection (complement reference 5):

Rahman, S. M., Perry, N., Müller, J. M., Kim, J., & Laratte, B. (2020). End-of-Life in industry 4.0: Ignored as before?. Resources, Conservation and Recycling, 154, 104539. Mastos, T. D., Nizamis, A., Vafeiadis, T., Alexopoulos, N., Ntinas, C., Gkortzis, D., ... & Tzovaras, D. (2020). Industry 4.0 sustainable supply chains: An application of an IoT enabled scrap metal management solution. Journal of Cleaner Production, 269, 122377.   General concept: Xu, L. D., Xu, E. L., & Li, L. (2018). Industry 4.0: state of the art and future trends. International Journal of Production Research, 56(8), 2941-2962.   Additionally to the above stated references, about 10-15 journal or peer-reviewed proceedings should be added to better embed the results and discuss them with extant literature, also from the journal Sensors.

Author Response

Dear reviewer,

Thank you very much for the valuable points in order to improve the quality of our scientific article.

In the following we are going to address and discuss our improvement based on your inputs:

[ Review comment 1]: The title is quite vague and might be better specified to what is actually done in the article.

[Authors reply to comment 1]: The title has been changed to “Smart node networks orchestration: a new E2E approach for analysis and design for Agile 4.0 implementation” in order to better explain where we position with our research topic and the methodology developed to support SMEs using Agile framework for the adoption of I4.0 technology.

[ Review comment 2]: The introduction does not derive a clear research gap that this manuscript covers. What is the novelty compared to extant research and why do we need this paper? Further, no references are used.

[Authors reply to comment 2]: The introduction of the article has been incremented with more references to the state-of-the-art research, better describing actual gaps in standardization and problems in the adoption of I4.0 technologies in order to clarify why our paper contributes in supporting the adoption of IoT technologies, leveraging on Systems Engineering analysis swimlane.

[ Review comment 3]: The description of Industry 4.0 in section 2.1 is not complete, not based on enough references, and does not link well to the rest of the paper.

[Authors reply to comment 3]: Regarding the section 2.1 that concerns with Industry 4.0 general context, we really appreciated the suggestions and the references attached to the revision. In particular, we better described the barriers limiting the introduction of modern 4.0 IT solutions: that is really linked to the overall research project inside which we developed the two described use cases, project where we worked together with local institution to support the knowledge transfer of I4.0 to the employees of selected companies.

[ Review comment 4]: It must be better reasoned why this use case and method was chosen, how it is representative and generalizable for other use cases.

[Authors reply to comment 4]: At the beginning of chapter 4 we better describe the interviews performed to understand the needs of local SMEs to be eventually addressed with I4.0 technologies, and we describe two IoT use cases that for their design patterns are representative of really different needs resolved with a common approach, both on functional and physical level. In such chapter is also described the framework used to perform knowledge transfer from our research team to the companies.

[ Review comment 5]: The following section 4 must be structured better, guiding the reader what is done where. Further, the headings and subheadings used here must be unified, some of them are outside the numbering scheme.

[Authors reply to comment 5]: In the future research section, we better describe our goals in relation to the results, and in particular the gaps, of the current article. We focus on hard real-time applications (reason for researching on potentiality of 5g in IIOT domain), DevOps methodology with real embedded hardware in the loop in order to face the need of great speed with high reliability of the systems in a continuous verification with CI/CD pipelines.

[ Review comment 6]: A discussion with extant literature and what this paper contributes is almost completely absent. Future research must be better connected to the results, it seems to falls from the sky, while interesting as such. Section 6 is a sum-up, but does not cover contribution, practical implications, or limitations. The list of references is quite low, the article needs much more journal or peer-reviewed references to embed and discuss its results. Further, some references do not have the adequate referencing style (reference 1 is incomplete) and there are a lot of online reports and not peer-reviewed conferences, but too little academic references for an academic article. Please find several suggestions below that should be used to embed the article better in literature:

[Authors reply to comment 6]: We fully revised the reference to the current state of the art, improving the paper with a wider bibliography.

Reviewer 2 Report

Paper is good, and appropriate for sensors, the authors present a case study of IoT pilot deployment for industry 4.0. In general, is good but they use not common industrial protocols, for example, they should consider using OPC-UA from www.opcfoundation.org and Data Distribution Service (DDS) from www.dds-foundation.org for data flow in the local network, as both they support by default search and discovery of nodes and data streams. Finally, they should use more scientific references most of the references are web pages only. They could improve also the HMI description as it is a bit light in this section.

Author Response

Dear reviewer,

Thank you very much for the valuable points in order improve the quality of our scientific article.

We take the opportunity to thank you for the several inputs that really helped us to better contextualize the goal and the achievements of our research, probably not so much explicitly highlighted in the previous version of the article.

In the following we are going to address and discuss our improvement based on your inputs:

[ Review comments]:  Paper is good, and appropriate for sensors, the authors present a case study of IoT pilot deployment for industry 4.0. In general, is good but they use not common industrial protocols, for example, they should consider using OPC-UA from www.opcfoundation.org and Data Distribution Service (DDS) from www.dds-foundation.org for data flow in the local network, as both they support by default search and discovery of nodes and data streams. Finally, they should use more scientific references most of the references are web pages only. They could improve also the HMI description as it is a bit light in this section.

[Authors replies ]

1 – Regarding the IoT protocol we decided to use MQTT as peer-to-peer asynchronous protocol, since in the two use cases explained there were no critical time constraints and hard requests in terms of sample availability; as better described in the current release the focus was more on using solutions based on embedded cheap systems based on open hardware and software. For sure in our future research we’ll address and scale the approach also to different applications (mainly addressing Manufacturing 4.0) were hard real-time constraints and industrial standardization with DDS and/or OPC is valuable solution.

2 – As far as it concerns literature review we expanded in the first chapters, focusing on major barriers in the adoption of modern Industry 4.0 technologies: this topic is really linked to the overall research project inside which we developed the two described use cases.

3 – The description of the HMI was improved introducing more details on the UX study that carried to the design of the website map and interface and also on the methodology of validation with unexpert alpha testers selected among the employees of a company.

Reviewer 3 Report

This paper presents 2 Industrial use cases for the application of smart node sensor orchestration. My concerns about the proposed solutions are presented below.

Introduction -- it is not clear the topic of the paper. In the abstract, you said that "n the use case section, the prototypes developed as proof of concept and the KPIs used for the system validation are described to provide a concrete solution overview"  

What are the better solutions proposed than those presented in the literature? what are the novelties introduced in the 2 proposed solutions?

The proposed architecture from figure 11 uses wireless IoT nodes. In my opinion, wireless communication is not suitable for industrial areas. In these areas there are high RF interference environments. For this reason, there are used wired fieldbuses such as CanOpen, Ethercat, Modbus etc. 

Furthermore, MQTT middleware is not used in the industrial environment. There are other middleware systems, such as DDS that provide support for real time --- very important in industrial applications.

Author Response

Dear reviewer,

We take the opportunity to thank you for the several inputs that really helped us to better contextualize the goal and the achievements of our research, probably not so much explicitly highlighted in the previous version of the article.

In the following we are going to address and discuss our improvement based on your inputs:

[ Review comment]

This paper presents 2 Industrial use cases for the application of smart node sensor orchestration. My concerns about the proposed solutions are presented below.

Introduction -- it is not clear the topic of the paper. In the abstract, you said that "n the use case section, the prototypes developed as proof of concept and the KPIs used for the system validation are described to provide a concrete solution overview"  

What are the better solutions proposed than those presented in the literature? what are the novelties introduced in the 2 proposed solutions?

The proposed architecture from figure 11 uses wireless IoT nodes. In my opinion, wireless communication is not suitable for industrial areas. In these areas there are high RF interference environments. For this reason, there are used wired fieldbuses such as CanOpen, Ethercat, Modbus etc. 

Furthermore, MQTT middleware is not used in the industrial environment. There are other middleware systems, such as DDS that provide support for real time --- very important in industrial applications.

[Authors reply to comment]

1 – Regarding the novelty of the paper, the focus was more on the analysis approach and the E2E framework for design and implementation, in order to support SMEs in the adoption of distributed systems via IoT technologies harmonizing and creating a possible flow starting from requirements and arriving to validation phase. We better explained and highlighted this fact in the chapter 4, where we want to support the introduction of I4.0 technologies in a simple way, also in enterprises where such adoption is limited due to lack of domain knowledge.

2 – The two use cases presented had the requirements of wireless connectivity, for this reason we focused on this hardware architecture for sensor nodes based on wifi modules. As far as concerns the software architecture, it leverages on web services exchanging information via internet protocols and is scalable also to nodes where wireless communication could be difficult due to RF noise, this is achievable for instance creating client-server M2M data communication with IoT node present in a wired LAN.

3 – Regarding the IoT protocol we decided to use MQTT as peer-to-peer asynchronous protocol, since in the two use cases explained there were no critical time constraints and hard requests in terms of sample availability as better described in the current release the focus was more on using solutions based on embedded cheap systems based on open hardware and software. For sure in our future research, we’ll address and scale the approach also to different applications (mainly addressing Manufacturing 4.0) were hard real-time constraints and industrial standardization with DDS and/or OPC is valuable solution.

Round 2

Reviewer 1 Report

While several points have been solved well, the following aspects from the first review round have not been considered:

  • The introduction [...] no references are used.
  • A discussion with extant literature and what this paper contributes is almost completely absent. The same applies for practical implications and limitations that are still quite short.

The following suggested references (or alternative ones) have not been cited, while there are still no references from the journal sensors:

  • Rahman, S. M., Perry, N., Müller, J. M., Kim, J., & Laratte, B. (2020). End-of-Life in industry 4.0: Ignored as before?. Resources, Conservation and Recycling, 154, 104539.
  • Mastos, T. D., Nizamis, A., Vafeiadis, T., Alexopoulos, N., Ntinas, C., Gkortzis, D., ... & Tzovaras, D. (2020). Industry 4.0 sustainable supply chains: An application of an IoT enabled scrap metal management solution. Journal of Cleaner Production, 269, 122377.  
  • No references from this journal are cited.

Author Response

Dear reviewer,

Thank you very much for the inputs that helped us to improve our scientific article.

In the following we are going to address and discuss our improvement:

[ Review comment 1]: The introduction [...] no references are used. A discussion with extant literature and what this paper contributes is almost completely absent. The same applies for practical implications and limitations that are still quite short.

[Authors reply to comment 1]: Now in the introduction there is a broader contextualization of our article with respect recent scientific state of the art; leveraging on peer reviewed papers, we introduced references to industrial cyber-physical systems in Industry 4.0. To improve such section, we inserted both the papers suggested by reviewers, papers of the journal and other external relevant references. Then, a focus to the entire value chain is provided, not only focusing on enabling technologies, but also discussing the concept of circular economy: concepts such as recycling, end of life, but also proactive diagnostic to improve the processes. In this way we better highlight the implications of cyber-physical systems that are not only affecting the product or service delivered to the end consumer. We describe with more attention how our paper is integrated in the discussion about solutions to tackle barriers that are preventing a wider adoption of I4.0 technologies in SMEs and how we are going to provide a contribute, both in terms of knowledge sharing by means of agile methodologies, and in terms of system analysis and technical design. Both in the introduction and in the conclusion sections, we better explain what the main limitations of the current research are (i.e., system without hard real-time constraints and high availability of messages, the necessity of structured devops processes, necessity of having low power devices in the loop for cheap solutions) in order to contextualize the selected future research topics.

Reviewer 3 Report

The paper was improved by the revision process.

Author Response

Dear reviewer,

thank you for appreciating the improvements made to the paper.

In the following we are going to discuss additional improvements, in order to enrich mainly the contextualization of our work in the current state of the art and main implications and limitations:

Now in the introduction there is a broader contextualization of our article with respect recent scientific state of the art; leveraging on peer reviewed papers, we introduced references to industrial cyber-physical systems in Industry 4.0. To improve such section, we inserted both the papers suggested by reviewers, papers of the journal and other external relevant references. Then, a focus to the entire value chain is provided, not only focusing on enabling technologies, but also discussing the concept of circular economy: concepts such as recycling, end of life, but also proactive diagnostic to improve the processes. In this way we better highlight the implications of cyber-physical systems that are not only affecting the product or service delivered to the end consumer. We describe with more attention how our paper is integrated in the discussion about solutions to tackle barriers that are preventing a wider adoption of I4.0 technologies in SMEs and how we are going to provide a contribute, both in terms of knowledge sharing by means of agile methodologies, and in terms of system analysis and technical design. Both in the introduction and in the conclusion sections, we better explain what the main limitations of the current research are (i.e., system without hard real-time constraints and high availability of messages, the necessity of structured devops processes, necessity of having low power devices in the loop for cheap solutions) in order to contextualize the selected future research topics.